# Association between Udder and Quarter Level Indicators and Milk Somatic Cell Count in Automatic Milking Systems

**DOI:** 10.3390/ani11123485

**Published:** 2021-12-07

**Authors:** Maddalena Zucali, Luciana Bava, Alberto Tamburini, Giulia Gislon, Anna Sandrucci

**Affiliations:** Dipartimento di Scienze Agrarie e Ambientali, Università degli Studi di Milano, 20133 Milano, Italy; maddalena.zucali@unimi.it (M.Z.); alberto.tamburini@unimi.it (A.T.); giulia.gislon@unimi.it (G.G.); anna.sandrucci@unimi.it (A.S.)

**Keywords:** automatic milking system, dairy cows, somatic cell count, milk electrical conductivity

## Abstract

**Simple Summary:**

In dairy cattle herds milked by automatic systems, the absence of a human milker originates the need for control systems to monitor the milking process and cow conditions. Modern milking robots are equipped with a lot of sensors that, at each milking (2.5–3 times a day), record data on milk yield and quality, milking efficiency, cow welfare, and health with particular focus to udder conditions. Mastitis is one of the most frequent and serious diseases of dairy cow that negatively affects milk quality and yield, reduces animal welfare, and often implies the use of antimicrobial drugs. At the moment, the alerting systems for mastitis risk is generally based on monitoring milk electrical conductivity, color, and/or temperature, but these indicators have limited reliability. Other information gathered by automatic sensors, already implemented in commercial robots, could be useful to early detect mastitis. Using a multivariate approach, our study showed that the deviations over time of milk electrical conductivity, milk yield, and milk flow of single quarters in comparison with the whole udder are potential indicators, alone or in combination, for altered udder conditions. The results could be useful for the development of new algorithms more effective in the early detection of mastitis.

**Abstract:**

Automatic Milking Systems (AMS) record a lot of information, at udder and quarter level, which can be useful for improving the early detection of altered udder health conditions. A total of 752,000 records from 1003 lactating cows milked with two types of AMS in four farms were processed with the aim of identifying new indicators, starting from the variables provided by the AMS, useful to predict the risk of high milk somatic cell count (SCC). Considering the temporal pattern, the quarter vs. udder percentage difference in milk electrical conductivity showed an increase in the fourteen days preceding an official milk control higher than 300,000 SCC/mL. Similarly, deviations over time in quarter vs. udder milk yield, average milk flow, and milking time emerged as potential indicators for high SCC. The Logistic Analysis showed that Milk Production Rate (kg/h) and the within-cow within-milking percentage variations of single quarter vs. udder milk electrical conductivity, milk yield, and average milk flow are all risk factors for high milk SCC. The result suggests that these variables, alone or in combination, and their progression over time could be used to improve the early prediction of risk situations for udder health in AMS milked herds.

## 1. Introduction

Automated Milking Systems (AMS) are spreading rapidly in dairy cattle farms; at the beginning of the 2000s there were about 1250 milking robots around the world [1], but in 2017, a total of about 38,000 AMS units installed globally was reported [2].

The spread of AMS and the parallel reduction in human presence have generated the need to incorporate into the system automatic sensors for monitoring milk production, milking efficiency, and animal conditions. AMS is integrated with a variety of sensing equipment for the detection of multiple indicators during the milking process (e.g., milk yield, milk flow rate, milk electrical conductivity, incomplete milking, refusals, and concentrate intake). In recent years, a number of sensors has been developed for monitoring additional variables, for example, milk composition, milk temperature, somatic cell count, cow live weight, body condition score, lameness, etc. With a number of daily milkings at herd level of about 2.5, the AMS enables farmers to collect a huge amount of real time data on individual cows and their performances, in many cases at a quarter level, which allows for a detail in the monitoring never achieved so far.

The big data generated by the multiple sensors implemented in the AMS can be statistically managed for the development of algorithms for the real-time detection of changes that can enable the farmer to implement timely and optimized management actions. AMS can be considered not only as an alternative to traditional milking systems but also as a new and general approach to manage dairy herd health and production efficiency [3]. In particular, researchers have been working to integrate multiple data, mainly related to milk quality and ejection at quarter and udder level, for overcoming the limits of the indicators currently used in mastitis detection [4,5].

Mastitis is the most important disease that affects dairy herds both in terms of prevalence and economic costs: it causes production losses, milk quality alterations, medication and extra labour costs, discarded milk, and early culling of animals [6]. Moreover, mastitis has long-term detrimental effects on animal welfare [7] and increases the environmental impact of milk production [8]. Lastly, the use of antimicrobial agents for the prevention and control of mastitis is related to the risk of the development of antimicrobial-resistant bacteria, which is a global concern for human health and food safety [9]. In clinical mastitis, the cow has evident physical manifestations, which can be easily noticed by the milker, favouring prompt intervention. The diagnosis is much more difficult for subclinical mastitis, which is the predominant form of bovine mastitis, and is characterised by the absence of visible alterations in either the udder or in the milk. Microbiological examination of milk is considered to be the gold standard for diagnosing bovine mastitis, but costs and the time needed for culturing limit its use. Among the indirect methods for diagnosing bovine mastitis, automatic somatic cell counting (SCC) is considered the most reliable but individual somatic cell count is usually provided on a monthly basis by the National Breeders’ Association controls and the time intervals are too long for early diagnosis. Both conventional and automatic milking systems generally do not measure somatic cell count: their alerting systems are based on milk electrical conductivity and milk colour [4,5]. Khatun et al. [10] suggested that milk electrical conductivity alone is not likely to achieve the required sensitivity and specificity and that mastitis detection can be improved by using multiple measurements. At the moment some questions are open on the possibility and the reliability of using automatically collected data by AMS to assess udder health problems in dairy cow.

In this context, the aim of the study was to examine the variables automatically recorded by the AMS sensors, and some new derivative indicators, studying their association, changes over time, and ability to predict the risk of increasing milk somatic cell count.

## 2. Materials and Methods

### 2.1. Dataset

Four dairy cattle farms located in Lombardy (northern Italy), equipped with automatic milking systems of two different brands (VMS, DeLaval, Tumba, Sweden and AMS, Lely Industries NV, Maassluis, the Netherlands), were involved in the study. DeLaval’s Milking Robots is labelled “A AMS” and Lely’s Milking Robots is labelled “B AMS”. The dataset included information recorded by the AMSs in the period 2016-2019. A total of 602,000 records of 720 cows from the two farms using type A AMS, and 150,000 records of 283 cows from the two farms using type B AMS were collected. All the cows were kept in a free-stall housing, without access to pasture; lying cubicles were bedded with straw and sawdust. Cows were fed a partial mixed ration integrated by concentrate in the AMS stall. In Table 1, the main characteristics of the farms are reported (Table 1).

From Table 1, it can be noted that between 2016 and 2019 three out of four farms increased milk yield per cows by approximately 10 to 16% and all the farms improved cow fertility reducing days open by 5 to 21%.

The two types of AMSs slightly differed for the data recorded and made available. In particular, B AMS registered a lot of variables on milking efficiency such as number of daily refusals, failures, and downtimes. Moreover, B AMS captured indicators of udder milk composition (fat, protein, and lactose) and somatic cell count by sensors, based on NIR spectroscopy and viscosity, respectively. On the contrary, A AMS made available a lot of information per quarter such as milk yield and milk flow. Both systems measured milk electrical conductivity per quarter, but in the case of B AMS, conductivity was expressed as a score out of 100, instead of mS/cm. Consequently, variables studied were partly different for the two AMSs. Table 2 reports the variables analysed, their units and acronym.

In addition to the information obtained by the AMSs, milk production and quality data from individual monthly official milk controls performed by the National Breeders’ Association were included in the dataset. Milk samples were analysed for fat and protein contents (MilkoScan FT6000; Foss Analytical A/S, Hillerod, Denmark) and milk somatic cell count (Fossomatic). Somatic cell count was log-transformed as Linear Score (LS), using the equation suggested by Wiggans and Shook [11]: LS = log_2_ ((somatic cell count/mL)/12,500).

Individual cow data as days in milk (DIM, d), stage of lactation (STAGE: <100; 100–200; >200 DIM) and number of lactation (nLACT: 1, 2, ≥3) were included in the dataset.

Each individual cow record was classified as a function of somatic cell count obtained from official controls as LOW (≤300,000 cell/mL) or HIGH (>300,000 cell/mL). Moreover, cows with more than 80% of the official milk analyses characterized by LSc ≤ 5 (SCCc ≤ 100,000 cells/mL) were classified as low SCC (LSCC); cows with more than 80% of the official milk analyses characterized by LSc > 5 (SCCc > 100,000 cells/mL) were classified as high SCC (HSCC); in all the other cases cows were classified as fluctuating SCCc (FSCC).

A number of new additional variables were calculated starting from the data recorded by the AMSs: for all the herds milk production rate at udder level, as rate of milk secretion per hour (MPR = YM/MI; kg/h); for the herds milked with A AMS, the within-cow within-milking percentage differences between individual quarters and the average of the four quarters of the minimum milk yield (ΔYq, %; ΔYq = (minYq − meanYq)/meanYq * 100), the minimum average milk flow (ΔAMFq, %), the maximum milk electrical conductivity (ΔECq, %); for the herds milked with B AMS, the within-cow within-milking percentage differences between the individual quarters and the average of the four quarters of the minimum milking times (ΔTq, %); and the maximum milk conductivity score (ΔECSq, %).

### 2.2. Statistical Analyses

The whole dataset was analysed using SAS software (Version 9.4, SAS Institute, Inc., Cary, NC, USA, 2012) always separately for the two types of AMS. Descriptive statistics and Pearson’s correlation analyses were obtained using Proc MEANS, Proc FREQ and Proc CORR. Differences between front and rear quarters and between right and left ones were evaluated using a GLM procedures.

A GLM analysis was performed to test the differences among the three SCC cow classes in terms of ΔECq, ΔYq, ΔAMFq for the A AMS herds and in terms of ΔECSq and ΔTq for the B AMS herds. Number of lactation, stage of lactation, and farm effects were included in the models.

Two types of multivariate analyses were performed: a Principal Component Analysis (PROC PRINCOMP), and a Logistic Analysis (PROC LOGISTIC).

Through the Principal Component Analyses, the relationships among data from official controls, data recorded by the AMSs, and the additional variables calculated were studied for each AMS type.

Class variables were created to perform the Logistic Analyses with the aim of identifying the main categorical variables associated with high level of milk SCC in the official controls (SCC > 300,000 somatic cells/mL). Dependent variables were MPR (<1.5; 1.5 ÷ 1.7; >1.7 kg/h), ΔECq (≤2.5%; >2.5%), ΔAMFq (≤−15.0%; >−15.0%), ΔYq (≤−24.0%; >−24.0%), ΔECSq (≤2.2%; >2.2%), and ΔTq (≤−25.0%; >−25.0%).

In the Logistic Analyses, variables or combinations of variables (interaction terms) were excluded through a stepwise backward multiple regression method based on a 0.2 significance level. The results of the analyses built, for each AMS type, a final model including variables (risk factors) that were significantly associated with SCC > 300,000/mL. The final models were described in terms of odds ratios and 95% confidence intervals.

## 3. Results and Discussion

### 3.1. Descriptive Statistics

The general descriptive statistics of the herds milked with the two AMS (A and B) are reported in Table 3. The main characteristics of the two couples of herds were similar in terms of herd traits, milk production, and composition. In the A AMS milked herds, LS was higher than in the herds milked with B AMS, suggesting slightly poorer udder health conditions. Herds milked using A AMS showed lower milking efficiency with lower number of milkings per day, longer interval between milkings and slightly lower milk production rate, MPR. The low average value of daily milkings of the A AMS herds is probably related to farm management problems and to the production disciplinary of the Protection Consortium Grana Padano PDO that, until the end of 2019, imposed a maximum number of daily milkings per cow not exceeding two.

The A AMS provides a lot of information on milk ejection at a quarter level. In particular average milk flow (1.13 ± 0.32 kg/min), peak milk flow (1.61 ± 0.38 kg/min), milk yield per milking (3.63 ± 1.22 kg/milking), and milk electrical conductivity (4.12 ± 0.35 mS/cm). According to Juozaitienė et al. [12], the average milk electrical conductivity (EC) of healthy cows milked with AM systems ranges from 4.6 to 5.8 mS/cm. Norberg [13] reported that the maximum milk electrical conductivity of healthy quarters within cow and within milking was 4.87 mS/cm, whereas subclinically and clinically infected quarters had maximum values of 5.37 mS/cm and 6.44 mS/cm, respectively. As pointed out by Penry et al. [14] and Kathun et al. [4], cisternal milk has a higher EC compared with alveolar milk. Once alveolar milk ejection starts, EC decreases as a consequence of mixing alveolar and cisternal milk. This is a confounding factor that influences milk EC recorded by automatic sensors and makes it impossible to fix a threshold value as an early signal of altered quarter health.

The B AMSs, in addition to some information at a quarter level, provide a lot of information on milking efficiency and indications about milk components and somatic cell count at udder level from the AMS sensors. The most important variables registered at quarter level were milking time (2.70 ± 1.17 min) and milk electrical conductivity expressed as score (69.2 ± 3.61). At udder level, average milk flow was 3.57 ± 1.10 kg/min and indications of whole udder milk components from sensors were 3.9 ± 0.83%, 3.4 ± 0.21%, and 5.00 ± 0.11% for milk fat, protein, and lactose, respectively. Milk somatic cell count from AMS sensors was 225,695 ± 451,540 cell/mL and the milk temperature was on average 39.1 ± 0.64 °C.

For the herds milked with A AMS, the average % difference within-cow and within-milking of milk yield of the least productive individual quarters compared to the mean of the four quarters (ΔYq) was −24.2 ± 15.6%; the average % difference in average quarter milk flow (ΔAMFq) was −16.9 ± 10.7%; and the average % difference in quarter milk conductivity (ΔECq) was 3.73 ± 4.24%.

For the herds milked with B AMS, the average % difference in milking time of individual quarters compared to the average of the four quarters (ΔMTq) was −25.1 ± 16.4%, and the average difference in electrical conductivity score (ΔECSq) was 2.22 ± 3.14%.

### 3.2. Quarter Position

In A AMS herds, rear quarters showed significantly higher milk yield in comparison to front quarters (4.01 vs. 3.03 kg/milking; P < 0.0001; 57% vs. 43%), higher average milk flows (1.15 vs. 1.07 kg/min; P < 0.0001), and higher peak milk flows (1.63 vs. 1.52 kg/min; P < 0.0001) in agreement with previous studies [5,15,16,17,18]. In B AMS herds, rear quarters showed significantly higher milking time than front ones (3.11 vs. 2.56 min; P < 0.0001). Milk electrical conductivity was significantly different between rear and front quarters in all the four herds but the differences were very small, inconsistent between the herds milked with the two AMS types, and not of biological relevance.

In comparison to the left quarters, right quarters produced significantly more milk (3.57 vs. 3.48 kg/milking; P < 0.0001) and had slightly higher average and peak milk flows in A AMS herds. In B AMS herds, right quarters showed significantly lower milking time (2.81 vs. 2.85 min; P < 0.0001). Differences in milk ejection among left and right quarters were noted also by Tancin et al. [19]; this could be a consequence of the teat cup attachment order and the different levels of circulating oxytocin; although, some authors have denied variations in the levels of oxytocin and alteration of milk ejection due to the attachment sequence and possible delays in AMS [20,21].

### 3.3. Pearson’s Correlation Analyses

Pearson’s correlation coefficients are shown in Table 4 and Table 5. In the A AMS herds (Table 3) daily milk yield (from official controls, Yc) was positively correlated with MPR (r = 0.42) and negatively with milking interval (MI; r = −0.30). Milk electrical conductivity at quarter level (ECq) did not show any correlation with LSc at udder level (r = 0.03 ÷ 0.09 for the different quarters) but it was positively correlated with official daily milk yield (Yc; r = 0.20 ÷ 0.27 for the different quarters; data not shown in the table). Juozaitienė et al. [12] reported a polynomial relationship between udder milk electrical conductivity and milk yield: the best productivity was observed in the milk electrical conductivity class of 4.5-5.5 mS/cm. MPR showed a negative correlation with milking interval (MI; r = −0.39), in agreement with the findings of Penry et al. [16], who observed an MPR decrease of 2% per hour when increasing the MI for multiparous cows and 1.5% per hour for primiparous ones; although, in that case, MPR was determined at quarter level.

Lyons et al. [22] found a non-linear association between MPR and MI, with reduced MPR when the milking interval was more than 16 h. Average milk flow and peak milk flow at quarter level were highly positively correlated (r = 0.95 ÷ 0.96 for the different quarters; data not shown in the table) as observed by Penry et al. [23]. Although there was not any correlation between milk electrical conductivity of single quarters (ECq) and LSc, the % difference in electrical conductivity between quarter and udder (ΔECq) was positively correlated to LSc (r = 0.26). ΔECq was also negatively related to the % difference between quarter and udder of milk yield (ΔYq; r = −0.39) and average milk flow (ΔAMFq; r = −0.26).

In the B AMS herds (Table 5), similar to what has been observed in A AMS herds, daily milk yield (from official controls, Yc) was highly positively correlated with MPR (r = 0.85) and negatively correlated with the duration of MI (r = −0.41).

Milk electrical conductivity score of each quarter (ECSq) was negatively correlated with lactose indication (Lams; r = −0.76 ÷ −0.77 for the different quarters), but similar to what has been observed in A AMS herds, it did not show any correlation with LSc (r = 0.04 ÷ 0.07 for the different quarters) and SCCams (r = 0.11 ÷ 0.14 for the different quarters; data not shown in the table). Number of daily milkings (nM) was positively correlated with MPR (r = 0.50) and negatively correlated with MI (r = −0.70). MPR showed a positive correlation with number of lactation (nL; r = 0.33) and a negative correlation with MI (r = −0.48), similar to what was seen in the A AMS and in agreement the findings of other authors [16,22].

Correlation coefficients between milk components from official controls and indications provided by the B AMS sensors were 0.45 and 0.41 for milk fat and milk protein percentages, respectively. Somatic cell count provided by the B AMS (SCCams) showed a positive correlation with LSc (r = 0.38). The r coefficient values for milk component and SCC show a medium degree of correlation between the results provided by the AMS and the lab. This may depend on a limit of the AMS sensors but also on the fact that the official control provides only monthly values that may not necessarily be representative of the trend.

Although there was not any correlation among milk electrical conductivity score of single quarters (ECSq) and both LSc and SCCams, the % difference in milk electrical conductivity score between single quarters and udder (ΔECSq) was positively correlated to LSc (r = 0.27) and SCCams (r = 0.44).

### 3.4. Principal Component Analyses

The results obtained from the PCA are plotted in Figure 1a,b AMS herds.

The PCA on A AMS herds’ data (Figure 1a) confirmed the positive association among daily milk yield (from both AMS recordings, Yams, and official controls, Yc) and MPR: they are in the same quadrant and very close each other on the axis 1 that explains a good portion of the variance. Milk yield and MPR are in the opposite quadrant of milking interval (MI) in agreement with the findings of Penry et al. [16] and Lyons et al. [22]. As stated by Penry et al. [16], MPR can be considered a more appropriate way to measure milk yield in AMS where milking is voluntary and the MI is not uniform within or across days. On the other hand, milking interval (MI) is close to DIM on the axis 1. Penry et al. [16] reported a growth in MI with increasing lactation days: in multiparous cows, from 30 to 300 DIM, the average MI rose from approximately 8 to 10.2 h.

The analysis confirmed the positive association among LSc and % difference in milk conductivity (ΔECq) of single quarters vs. udder. These variables are in the opposite quadrant of the % difference in milk yield (ΔYq) and % difference in average milk flow (ΔAMFq) of single quarters vs. the whole udder.

Similar results were obtained from the PCA on B AMS data (Figure 1b). Daily milk yield (both from AMS recordings, Yams, and from the official controls, Yc), number of daily milkings (nM) and MPR are in the same quadrant. They are close each other on the axis 1 that explains 29.7% of the variance. The analysis confirmed the positive association among LSc, SCCams and % difference in milk conductivity score (ΔECSq) of single quarters compared to the whole udder. These variables are in the opposite quadrant of % milking time difference (ΔTq).

The indication of somatic cell count from AMS (SCCams) and linear score from official controls are very close on axis 1. Similar closeness emerges for milk fat and protein indications from AMS and milk fat and protein from official controls. These results suggest that the sensors for milk quality currently installed on many AMSs could be considered reliable enough for these parameters.

The results from the PCA suggest that the distancing of a single quarter from the condition of the whole udder in terms of milk electrical conductivity (ΔECq or ΔECSq), milk yield (ΔYq), average milk flow (ΔAMFq), and milking time (ΔTq) is well related to SCC and LS and can be useful as indicators of udder health problems.

### 3.5. GLM Analyses

In the A AMS herds, % difference milk electrical conductivity of quarter vs. the whole udder (ΔECq) was significantly influenced (P < 0.001) by the SCC cow class (LSCC, FSCC, and HSCC, respectively), nLACT, DIM, and SCCc. Percentage difference in quarter vs. udder milk yield (ΔYq) was significantly affected (P < 0.001) by the SCC cow class (LSCC, FSCC, and HSCC, respectively), nLACT, DIM, and herd. Similarly, in the B AMS herds, % difference in quarter vs. udder milk conductivity score (ΔMCSq) was significantly influenced (P < 0.001) by the SCC cow class, nLACT, DIM, herd, and SCCc.

Figure 2 shows the trend of the % difference in milk electrical conductivity (ΔECq for A AMS herds and ΔECSq for B AMS ones) in the 14 days before the official milk controls performed by the breeders’ association. Cows characterized by different SCC level in their production history (LSCC, FSCC, and HSCC, respectively) showed different trends for the variables in approaching the official milk control. In particular, in A AMS herds (Figure 2B), ΔECq showed increasing values approaching the day of official milk control when the SCC at the control was higher than 300,000 cell/mL, either in cows characterized by constant high milk SCCc (HSCC) or in cows with fluctuating SCCc (FSCC). Similar findings were obtained in B AMS herds (Figure 3) for ΔECSq; although, the trends were less clear.

This is in agreement with the results of the PCA where increasing ΔECq, where associated to higher LS al the official control. This result suggests that the increase in difference between the milk EC of a single quarter and the mean values of the four quarters in the days before the official milk control could be a good indicator of the risk to have a result of high SCC and can be useful as an early predictor of altered mammary health.

In A AMS herds (Figure 4), % difference in quarter vs. udder milk yield also revealed decreasing trends approaching the day of milk control when the SCC value at the control was higher than 300,000 cells/mL, either in cows characterized by constant high milk SCC or in cows with fluctuating SCC.

Inzaghi et al. [5] noted that positive indications of clinical mastitis were apparent well before confirmation of visual signs both for quarter milk electrical conductivity and MPR: from 5 to 12 milking sessions prior to confirmation of clinical mastitis depending on the number of lactation. In the same study average milk flow (AMF) at quarter level showed a downward trend prior to clinical mastitis confirmation.

### 3.6. Logistic Analysis

Logistic analyses were performed to identify the main risk factors for having a somatic cell count at the official control (SCCc) higher than 300,000 cells/mL. Table 6 shows that in the A AMS herds, In addition to the influence of DIM and number of lactation, the most important factor for high SCC was the MPR class: if the MPR is <1.5 kg/h the risk to have high SCC (>300,000/mL) is 2.26 times higher than the case of MPR > 1.7 kg/h. This result suggests that when MPR is low, for reducing milk yield or increasing milking interval, an udder health problem may be present. Inzaghi et al. [5] reported that quarters affected by clinical mastitis had consistently lower MPR than uninfected quarters. The reduction in MPR at udder level is probably an indicator of both the direct effect of disease in the infected quarter, as well as a systemic effect resulting in MPR reduction in healthy quarters of the same udder due to decreased lying, ruminating, and drinking time, and lower feed intake for cows with clinical mastitis. The second most important factor is the % difference in quarter vs. udder milk electrical conductivity: when ΔECq is ≤ 2.5% the risk to have high SCC (>300,000/mL) is more than halved (0.45) compared with the case of ΔECq > 2.5%. This result confirms what emerged from the GLM: when the milk electrical conductivity of single quarter deviates excessively from the mean of the four quarters, udder health problems can be suspected. In a recent study, Kathun et al. [4] explored the ability of % variation of quarter milk electrical conductivity to predict the onset of clinical sign of mastitis, but they did not obtain satisfying levels of Sensitivity and Specificity.

The logistic analysis showed also that when milk yield of the single quarter sensibly diverges from the mean value of the four quarters (ΔYq ≤ −24%) the risk of having high SCCc (>300,000/mL) at the milk control increases by 1.18 times in comparison with the case of ΔYq > −24%. Lastly, a lower risk (0.80) of high SCCc is associated with % difference in quarter vs. udder average milk flow (ΔAMFq) higher than −15%.

The relationship between the risk of mastitis or high somatic cells and average or peak milk flow is controversial. Grindal et al. [24] suggested that cows with greater quarter peak milk flows are more susceptible to mastitis. Average and peak milk flow give different information, but as reported before, they are highly correlated. In a study on conventional milked herds, peak milk conductivity and LS increased (P < 0.01) from the second (3 to 4 kg/min) to the third class (>4 kg/min) of peak milk flow rate [25] On the contrary, according to Tancin et al. [19], quarters with high SCC (>500 × 10^3^ cells/mL) had lower peak flow rate compared with quarters with low SCC (<200 × 10^3^ cells/mL). For AMS herds, Hammer et al. [26] reported an increased risk of clinical mastitis associated with low average peak milk flow at quarter level (<1.0 kg/min) occurring 7 to 14 d before occurrence of mastitis. However, analysis of quarter peak milk flow 0 to 7 d before onset of clinical signs indicated the risk of mastitis was associated with both low- and high-average quarter peak milk flow. As suggested by Inzaghi et al. [5], a wider teat canal favours bacterial entry and may slow down the refolding of the teat canal in the few minutes postmilking. On the contrary, the reduction in milk yield as a consequence of emerging mastitis can have the effect of reducing peak and average milk flow.

In the B AMS herds (Table 7), besides the effects of DIM and number of lactation, the most important factor for high SCCc was the indication for SCC from the sensors of AMS (SCCams): the risk of having high somatic cell count at the official milk control (SCCc) is about one-tenth when SCCams is ≤300,000 cell/mL than in the case of SCCams > 300,000 cell/mL. The second most important factor affecting the risk of SCCc higher than 300,000 cell/mL at the milk control is % difference in quarter vs. udder milk electrical conductivity score: if ΔECSq is <2.2% the risk to have high SCCc (>300,000/mL) is about halved (0.56) in comparison with the case of ΔECSq > 2.2%. Moreover, the analysis confirmed that when MPR is <1.5 kg/h the risk to have high SCCc (>300,000/mL) is higher (1.59 times) than the case of MPR > 1.7 kg/h. Lastly, lower protein indication from AMS sensors (Pams) is associated to lower risk of high SCC at the official control (SCCc).

## 4. Conclusions

The big data generated by the multiple sensors implemented in the automatic milking systems can be statistically managed to develop algorithms for the real-time detection of changes that may be indicative of mastitis or impaired health and welfare conditions of dairy cows. The results of the study showed that some variables automatically recorded by the AMSs at a quarter level can be used as good indicators of high SCC risk, when the deviation among the single quarters and the average of the four quarters is considered. Looking at the temporal patterns, the quarter vs. udder percentage difference in milk electrical conductivity showed an increase in the fourteen days preceding an official milk control higher than 300,000 cells/mL. Similarly, deviations over time in quarter vs. udder milk yield, average milk flow, and milking time emerged as potential indicators for high somatic cell count. The Logistic Analysis showed that Milk Production Rate (kg/h) and the within-cow within-milking percentage variations of single quarter vs. udder milk electrical conductivity, milk yield, and average milk flow are all risk factors for high milk somatic cell count.

The result suggests that these variables, alone or in combination, and their progression over time, could be used to improve the early prediction of risk situations for udder health in AMS milked herds.

## Figures and Tables

**Figure 1 animals-11-03485-f001:**
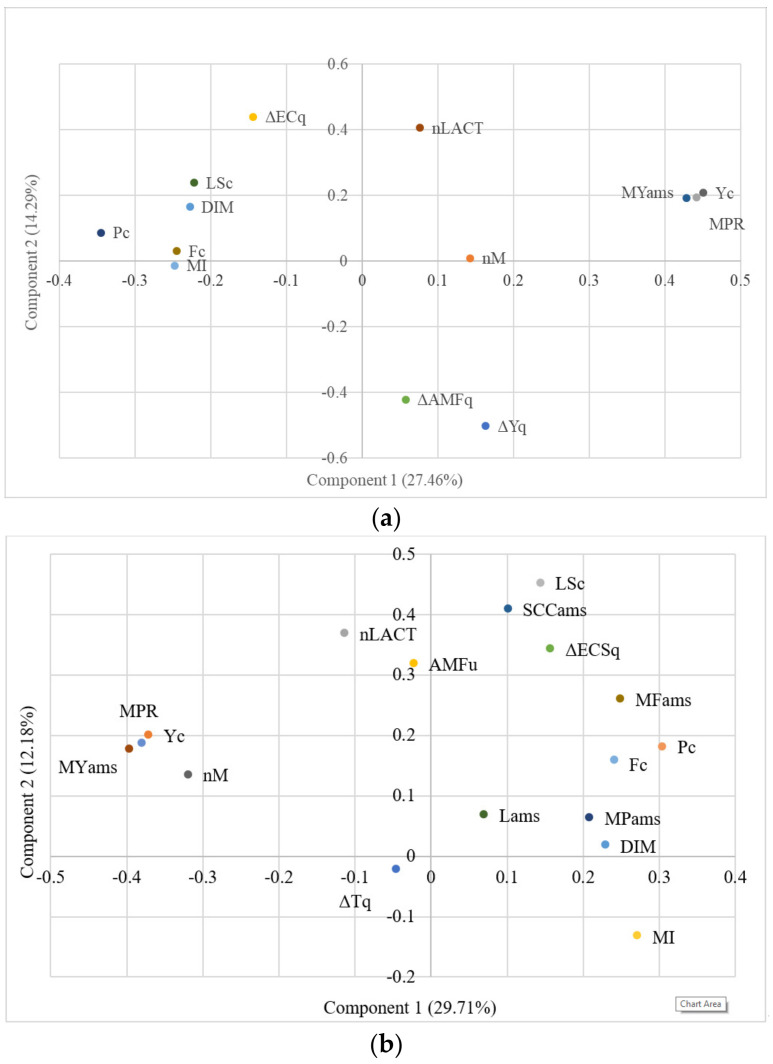
Principal Component Analyses: (**a**) A AMS herds—nLACT = number of lactation; DIM = days in milk; Yc = daily milk yield from official controls, Fc = milk fat percentage from official controls; Pc = milk protein percentage from official controls; LSc = Linear Score from somatic cell count determined by official controls; YMams = milk yield per milking from AMS; nM = number of daily milkings; MI = milking interval; Yams = daily milk yield from AMS; MPR = milk production rate; ΔYq = % difference quarter vs. udder milk yield; ΔAMFq = % difference quarter vs. udder average milk flow; ΔECq = % difference quarter vs. udder milk electrical conductivity; (**b**) B AMS herds—nLACT = number of lactation; DIM = days in milk; Yc = daily milk yield from official controls, Fc = milk fat percentage from official controls; Pc = milk protein percentage from official controls; LSc = Linear Score from Somatic cell count determined by official controls; YMams = milk yield per milking from AMS; nM = number of daily milkings; MI = milking interval; Yams = daily milk yield from AMS; MPR = milk production rate; AMFams = average udder milk flow from AMS; SCCams = SCC indication from AMS; Fams = milk fat percentage indication from AMS; Pams = milk protein percentage indication from AMS; Lams = milk lactose percentage indication from AMS; ΔTq = % difference quarter vs. udder milking time r; ΔECSq = % difference quarter vs. udder milk electrical conductivity.

**Figure 2 animals-11-03485-f002:**
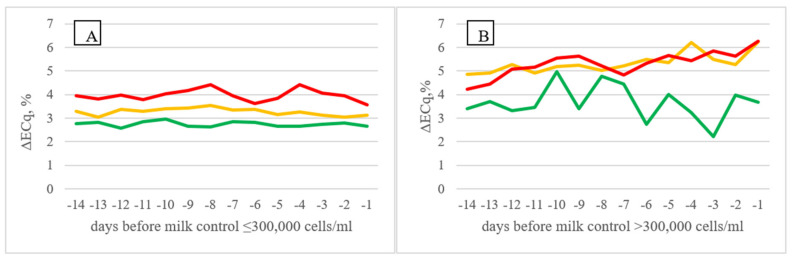
Difference in milk electrical conductivity of single quarters in (**A**) AMS herds, expressed as percentage of the average values within cow and within milking of the four quarters (Least Squares Means) in the 14 days before official milk control (≤300,000 cells/mL, (**A**); >300,000 cells/mL, (**B**)) on the basis of cow SCC classification (LSCC—green; FSCC—orange; HSCC—red).

**Figure 3 animals-11-03485-f003:**
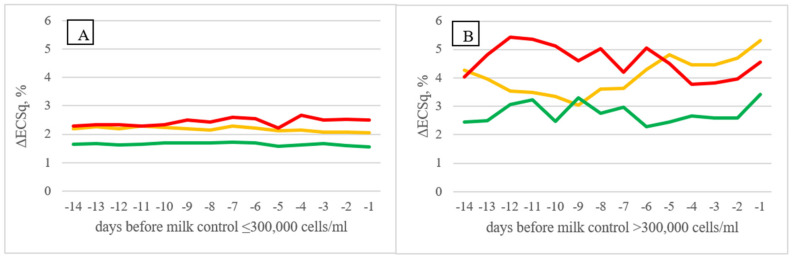
Difference in milk conductivity score of single quarters in B AMS herds expressed as percentage of the average values within cow and within milking of the four quarters (Least Squares Means) in the 14 days before official milk control (≤300,000 cells/mL, (**A**); >300,000 cells/mL, (**B**)) on the basis of cow SCC classification (LSCC—green; FSCC—orange; HSCC—red).

**Figure 4 animals-11-03485-f004:**
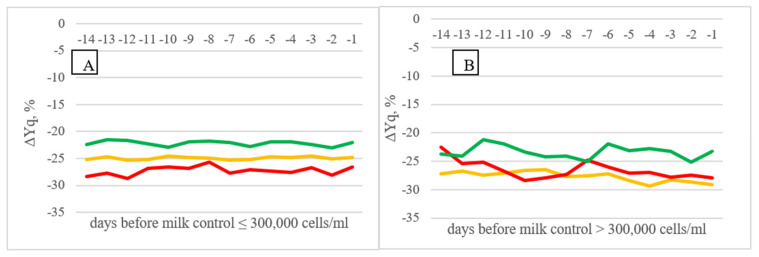
Difference in milk yield of single quarters in A AMS herds, expressed as percentage of average values within cow and within milking of the four quarters (Least Squares Means) in the 14 days before milk control (≤300,000 cells/mL, (**A**); >300,000 cells/mL, (**B**)) on the basis of cow SCC classification (LSCC—green; FSCC—orange; HSCC—red).

**Table 1 animals-11-03485-t001:** Main farm characteristics (no. cows, milk yield, and days open are obtained from the report of the official controls of the National Breeders’ Association for years 2016 and 2019, respectively).

Farm	Brand of AMS	AMS Model	Year of Installation	No. Robotic Units	Cow Traffic	No. Cows (2016–2019)	Breed	Average Milk Yield per Head (kg/y; 2016–2019)	Days Open (dd; 2016–2019)
1	DeLaval	VMS Classic	2017	1	free	38–40	Italian Holstein	10,974–10,452	142–128
2	DeLaval	VMS Classic	2019	2	free	130–145	Italian Holstein	10,178–11,793	197–156
3	Lely	A4	2013	1	free	61–66	Italian Holstein	11,839–13,067	149–129
4	Lely	A4	2017	3	free	161–208	Italian Holstein	10,410–11,944	134–127

**Table 2 animals-11-03485-t002:** Variables analysed for the two types of AMS, their acronyms and units.

Acronym	Units	Variable
BOTH AMSs		
Yc	kg/d	Daily milk yield from official controls
Fc	%	Milk fat from official controls
Pc	%	Milk protein from official controls
SCCc	cells/mL	Milk somatic cell count from official controls
LSc		Linear Score from SCC of official controls
YMams	kg	Udder milk yield per milking from AMS
MI	h	Milking interval
nM	no./d	Daily milkings
MPR	kg/h	Udder milk production rate
Yams	kg/d	Daily udder milk yield from AMS
A AMSs		
Yq	kg	Quarter milk yield per milking
AMFq	kg/min	Quarter average milk flow
PMFq	kg/min	Quarter peak milk flow
ECq	mS/cm	Quarter milk electrical conductivity
ΔYq	%	Difference in milk yield of single quarter vs. udder
ΔAMFq	%	Difference in average milk flow of single quarter vs. udder
ΔEC	%	Difference in milk electrical conductivity of single quarter vs. udder
B AMSs		
AMFams	min	Udder average milk flow from AMS
Fams	%	Udder milk fat indication from AMS
Pams	%	Udder milk protein indication from AMS
Lams	%	Udder milk lactose indication from AMS
SCCams	cells/mL	Udder milk somatic cell count indication from AMS
Tq	min	Quarter milking time
ECSq	%	Quarter milk electrical conductivity score
ΔTq	%	Difference in milking time of single quarter vs. udder
ΔECSq	%	Difference in electrical conductivity score of single quarter vs. udder

**Table 3 animals-11-03485-t003:** Descriptive statistics of the herds milked with the two automatic milking systems (A and B AMS).

		A AMS			B AMS		
Variable		*n*	Mean	SD	*n*	Mean	SD
Lactation-nLACT *	no. lactation	74,732	1.89	1.08	253,949	1.78	1
Days in milk-DIM *	d	74,732	156	108	253,618	148	92
Daily milk yield-Yc *	kg/d	74,732	37.5	8.78	253,460	39.9	8.94
Milk fat-Fc *	%	73,523	3.74	0.81	248,386	3.66	0.89
Milk protein-Pc *	%	73,523	3.32	0.35	248,386	3.39	0.36
Linear Score-LSc *		74,732	3.10	1.89	250,770	2.50	1.56
Milk yield per milking-YMams ^$^	kg/milking	144,208	14.0	4.69	602,471	13.0	3.85
Daily milkings—nM ^$^	no./d	144,208	1.88	0.85	602,468	3.05	0.69
Milking interval-MI ^$^	h	144,007	9.40	2.88	602,471	8.29	2.59
Daily milk yield-Yams ^$^	kg/d	134,441	37.5	10.7	602,471	39.4	9.6
Milk Production Rate-MPR	kg/h	144,208	1.56	0.75	602,471	1.64	0.44

* From official control records; ^$^ from AMS.

**Table 4 animals-11-03485-t004:** Pearson’s correlation coefficients among variables in the A AMS milked herds.

	nLACT	DIM	Yc	Fc	Pc	LSc	YM	nM	MI	Yams	MPR	ΔYq	ΔAMFq	ΔECq
nLACT	1													
DIM	0.01	1												
Yc	0.28	−0.29	1											
Fc	0.05	0.09	−0.33	1										
Pc	0.05	0.39	−0.46	0.37	1									
LSc	0.15	0.13	−0.20	0.26	0.27	1								
YMams	0.30	−0.12	0.40	−0.06	−0.18	−0.04	1							
nM	−0.05	−0.05	0.13	−0.08	0.18	−0.04	−0.28	1						
MI	0.12	0.13	−0.30	0.18	0.19	0.14	0.6	−0.41	1					
Yams	0.23	−0.25	0.77	−0.25	−0.38	−0.19	0.41	0.23	−0.27	1				
MPR	0.12	−0.16	0.42	−0.13	−0.22	−0.11	0.21	0.09	−0.39	0.42	1			
ΔYq	−0.15	−0.23	0.13	−0.07	−0.18	−0.14	0.09	0.03	−0.05	0.11	0.02	1		
ΔAMFq	−0.03	−0.08	0.01	−0.01	−0.02	0.00	0.06	0.00	0.02	0.03	−0.03	0.49	1	
ΔECq	0.14	0.07	−0.11	0.07	0.11	0.26	−0.03	−0.04	0.10	−0.12	0.02	−0.39	−0.26	1

All correlation coefficients were statistically significant (P < 0.001); nLACT = number of lactation; DIM = days in milk; Yc = daily milk yield from official controls, Fc = milk fat percentage from official controls; Pc = milk protein percentage from official controls; LSc = Linear Score from Somatic cell count determined by official controls; YMams = milk yield per milking from AMS; nM = number of daily milkings; MI = milking interval; Yams = daily milk yield from AMS; MPR = milk production rate; ΔYq = % difference quarter vs. udder milk yield; ΔAMFq = % difference quarter vs. udder average milk flow; ΔECq = % difference quarter vs. udder milk electrical conductivity.

**Table 5 animals-11-03485-t005:** Pearson’s correlation coefficients among variables in the B AMS milked herds.

	nLACT	DIM	Yc	Fc	Pc	LSc	YM	nM	MI	Yams	MPR	AMFams	SCCams	Fams	Pams	Lams
nLACT	1.00															
DIM	−0.08	1.00														
Yc	0.40	−0.45	1.00													
Fc	0.01	0.22	−0.37	1.00												
Pc	0.02	0.49	−0.43	0.50	1.00											
LSc	0.06	0.18	−0.16	0.27	0.27	1.00										
YMams	0.28	−0.08	0.37	−0.12	−0.14	−0.03	1.00									
nM	0.01	−0.27	0.50	-0.25	−0.25	−0.13	−0.25	1.00								
MI	0.00	0.26	−0.41	0.20	0.21	0.11	0.53	−0.70	1.00							
Yams	0.36	−0.40	0.93	−0.36	−0.41	−0.15	0.39	0.54	−0.46	1.00						
MPR	0.33	−0.37	0.85	−0.33	−0.37	−0.14	0.44	0.50	−0.48	0.92	1.00					
AMFams	0.18	0.08	0.21	−0.04	−0.03	0.14	0.28	0.00	0.05	0.22	0.24	1.00				
SCCams	0.11	0.07	−0.13	0.16	0.21	0.38	−0.07	−0.11	0.08	−0.16	−0.15	0.05	1.00			
Fams	0.02	0.33	−0.38	0.45	0.49	0.11	−0.19	−0.22	0.18	−0.4	−0.37	−0.09	0.15	1.00		
Pams	0.11	0.06	−0.2	0.32	0.41	0.06	−0.02	−0.23	0.19	−0.23	−0.22	−0.05	0.11	0.30	1.00	
Lams	−0.08	0.10	−0.07	0.09	0.09	−0.09	−0.07	0.07	−0.04	−0.06	−0.06	0.05	−0.09	0.05	0.01	1.00

All correlation coefficients were statistically significant (P < 0.001); nLACT = number of lactation; DIM = days in milk; Yc = daily milk yield from official controls, Fc = milk fat percentage from official controls; Pc = milk protein percentage from official controls; LSc = Linear Score from Somatic cell count determined by official controls; YMams = milk yield per milking from AMS; nM = number of daily milkings; MI = milking interval; Yams = daily milk yield from AMS; MPR = milk production rate; AMFams = average udder milk flow from AMS; SCCams = SCC indication from AMS; Fams = milk fat percentage indication from AMS; Pams = milk protein percentage indication from AMS; Lams = milk lactose percentage indication from AMS.

**Table 6 animals-11-03485-t006:** Logistic analysis of risk factors associated to >300,000/mL somatic cell count from official controls for A AMS herds.

Effect	Odds Ratio	95% Confidence Interval		P
DIM < 100 vs. >200	0.63	0.604	0.665	<0.0001
DIM 100–200 vs. >200	0.72	0.686	0.754	<0.0001
nLACT 1 vs. ≥3	0.46	0.439	0.485	<0.0001
nLACT 2 vs. ≥3	0.80	0.766	0.844	<0.0001
MPR < 1.5 vs. MPR > 1.7	2.26	2.145	2.376	<0.0001
MPR 1.5–1.7 vs. MPR > 1.7	1.31	1.243	1.377	<0.0001
ΔECq ≤ 2.5% vs. >2.5%	0.45	0.437	0.473	<0.0001
ΔYq ≤ −24% vs. >−24%	1.18	1.135	1.231	<0.0001
ΔAMFq ≤ −15% vs. >−15%	0.80	0.768	0.829	<0.0001

DIM = Days in milk; nLACT = Number of lactation; MPR = Milk Production Rate; ΔECq = % difference quarter vs. udder milk electrical conductivity; ΔYq = % difference quarter vs. udder milk yield; ΔAMFq = % difference quarter vs. udder average milk flow.

**Table 7 animals-11-03485-t007:** Logistic analysis of risk factors associated to >300,000/mL somatic cell count in the lab milk monthly control for B AMS herds.

Effect	Odds Ratio	95% Confidence Interval		P
DIM < 100 vs. >200	0.94	0.874	1.008	<0.0001
DIM 100–200 vs. >200	1.36	1.273	1.445	<0.0001
nLACT 1 vs. ≥3	0.41	0.380	0.439	<0.0001
Pams < 3.4 vs. >3.6	0.70	0.650	0.751	<0.0001
Pams 3.4–3.6 vs. >3.6	0.97	0.911	1.037	<0.0001
MPR < 1.5 vs. >1.7	1.59	1.478	1.718	<0.0001
ΔECSq ≤ 2.2% vs. >2.2%	0.56	0.528	0.590	<0.0001
SCCams ≤ 300,000 vs. >300,000	0.11	0.105	0.117	<0.0001

## Data Availability

The data presented in this study are available on request from the corresponding author.

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
