# Peer review of "Association between Udder and Quarter Level Indicators and Milk Somatic Cell Count in Automatic Milking Systems"

_animals, 2021, doi:10.3390/ani11123485_

Round 1
Reviewer 1 Report
Title:
I don't understand why the word "retrospective" is in the title? Do the data and results obtained only apply to the past? Rework the study title,
Introduction:
The introduction deals with AMS in very general terms. The work deals with milk somatic cells, but in the chapter Introduction there is no mention of their importance for dairy farming. The chapter needs to be edited.
Materials and Method:
A more detailed description of the monitored farms and their AMS must be made. Provide the following information in tabular form: Brands of AMS, exact type of AMS including year of produce and year of installation on farm and number of robotic unit on each farm, number and breed of dairy cows on each farm, average milk yield (kg) per cow per year for the monitored years 2016 – 2019, and occurrence of mastitis on each farm (%).
Results and Disscusion
The general problem is that there are a lot of results and they are confusing. Many tables and graphs. Some results are little discussed.
Table 2 - graphic error - line 3 is not complete.
Table 2 - the result of the Daily Milking parameter for A AMS is striking (1,88). Clear recommendation of AMS producers states that dairy cows must be milked at least twice per day in AMS! If not, the AMS will alert these dairy cows and the farm staff must take these dairy cows into the milking unit. Explain why this is not the case on the farms you have evaluated
I recommend a major revision of the manuscript
Author Response
Title:
I don't understand why the word "retrospective" is in the title? Do the data and results obtained only apply to the past? Rework the study title,
AU: thank you for your suggestion. We changed the title in “Association between udder and quarter level indicators and milk somatic cell count in automatic milking systems”
Introduction:
The introduction deals with AMS in very general terms. The work deals with milk somatic cells, but in the chapter Introduction there is no mention of their importance for dairy farming. The chapter needs to be edited.
AU: thank you for your suggestion, we improved introduction section adding some details on SCC and mastitis indicators commonly used.
Materials and Method:
A more detailed description of the monitored farms and their AMS must be made. Provide the following information in tabular form: Brands of AMS, exact type of AMS including year of produce and year of installation on farm and number of robotic unit on each farm, number and breed of dairy cows on each farm, average milk yield (kg) per cow per year for the monitored years 2016 – 2019, and occurrence of mastitis on each farm (%).
AU: Thank you for your suggestion, we added a table of the information requested
Results and Discussion
The general problem is that there are a lot of results and they are confusing. Many tables and graphs. Some results are little discussed.
AU: we provided to eliminate some secondary results
Table 2 - graphic error - line 3 is not complete.
AU: we fixed the graphic error, as suggested
Table 2 - the result of the Daily Milking parameter for A AMS is striking (1,88). Clear recommendation of AMS producers states that dairy cows must be milked at least twice per day in AMS! If not, the AMS will alert these dairy cows and the farm staff must take these dairy cows into the milking unit. Explain why this is not the case on the farms you have evaluated
AU: until the end of 2019 the production disciplinary of Grana Padano PDO imposed a maximum number of daily milkings per cow not exceeding two; therefore, in the farms equipped with robots whose milk was transformed into Grana Padano it was necessary to limit the number of daily milkings of the cows. We added a sentence in the text to explain the anomaly
I recommend a major revision of the manuscript
REV 2
The Introduction chapter is very limited in the references cited. There are only four of them. It is true that the authors took up various issues related to milking automation in the Introduction, but in my opinion they require more complete documentation in the form of quoting specialist literature. The literature in the field of research undertaken is very rich and it is worth using it in the form of citations. For example, the health issues of cows milked with a robotic milking and classic milking systems can be developed by citing selected studies, for example, "Assessment of dairy cow herd indices associated with different milking systems".
AU: thank you for your suggestion, we added some sentences and citations in the introduction section
In my opinion, after the review of the state of knowledge in the Introduction, it would be worth formulating a research problem. The research problem should result from a knowledge review indicating an insufficient state of knowledge in the area in question. This insufficient state of knowledge concerns in a given case - in my opinion - the possibility / reliability of using automatically collected data by AMS to assess health problems of dairy cows. Only the formulation of the research problem should be a premise to identify a gap in the current state of knowledge and to present the purpose of the research. In general, in the case of the formulated research goal, it would be worthwhile to simultaneously indicate a cognitive (scientific) goal and a utilitarian (useful) goal.
AU: we followed your suggestion and modified the end of introduction section.
On line 65, the authors wrote "(VMS, Delaval, Tumba, Sweden - A AMS)" and on line 66, "AMS, Lely Industries NV, Maassluis, the Netherlands - B AMS)". I am already omitting the fact that the poem with the Dutch company does not have an opening parenthesis at the beginning. However, it would be worthwhile to write "A AMS" and "B AMS" in a different way in the text of the article. Certainly, the authors' intention was to provide in lines 66 and 67 the designations of these two systems with "A AMS" and "B AMS". However, these markings are completely unreadable. I suggest you write a separate sentence that for research purposes, DeLaval's Milking Robots is labeled "A AMS" and Lely's Milking Robots is labeled "B AMS".
AU: thank you, we modified as suggested.
It's worth checking it out, but in my opinion, the name of the Swedish manufacturer is written as DeLaval, not Delaval (as the authors of the article wrote). It is therefore worth correcting the provisions of the company name in the article.
AU: thank you, we modified as suggested.
In my opinion, the description of the research material is not sufficient. If the research took into account SCC and other data related to the health and risk of cows, it would be advisable to indicate what type of laying bed was used on the farms. How often has the bedding material been replaced or refilled (in the case of straw / sawdust / sand) or cleaned. Were the cows kept in buildings with a loose housing system or a free-stall system? Was the lying area in the lying stalls (if the free-stall system) met the requirements of hygiene and how was this hygiene maintained (the employee removed the impurities?).
AU: as suggested by the reviewers we added a table (table 1) with main farm characteristics and some details on the housing.
I learned from the Materials and Methods chapter that the research included data from four farms, two for each company equipping farms with automatic milking systems. The authors also wrote how many cows the data came from and how much data there is, collected in the period 2016-2019. However, I did not find out how many milking robots were working on each farm. Were they one-stall milking robots? Theoretically, readers should know if they were one-stall AMS, if they are familiar with the companies' offerings. However, probably not everyone needs to know it, and therefore it would be worth confirming in the description in the Materials and Methods chapter. For me, the DeLaval VMS model is clearly identified. However, for the Lely milking robots included in the studies, more details need to be provided. Have there been a Lely Astronaut or A3 or A4 milking robot model on the farms? Please provide these details as they are important for the general description of the research material (study objects).
In the Materials and Methods chapter, the authors wrote that the data came from 720 cows (operated by the A AMS robot) and 283 cows (operated by the B AMS robot). And I would like to know the size of the herd milked with a milking robot on each farm under real conditions. What was the average herd size (± SD - standard deviation) milked with a milking robot on the farm and was it within the required range. For me, such information is the basic information presenting research objects, which in my opinion constitute a key element of description and comparison with other studies, as well as the repeatability of the authors' own research.
AU: we added this type of information in table 1
If the authors consider in-line and on-line sensors, it would be worth writing where they are exactly in the milking robot installation. Such details are not known to all readers, so - in my opinion - it needs to be described in more detail.
AU: in our opinion the detailed description of the sensors and their positioning risks to lengthen and burden the text and it is not crucial for the aims of the paper. We provided to delete from the manuscript the expression on-line and in-line
In subsection 2.2 (Statistical analyses) it would be worth mentioning the use of descriptive statistics to present the characteristics of cows / herds / farms. The results of descriptive statistics are presented in subsection 3.1, but it should be mentioned earlier, i.e. in subsection 2.2.
AU: If we are not wrong we mentioned these at lines 175-176
In the commentary to the result data summarized in Table 2, the Authors mentioned LS (line: 167), but such a concept (LS) is not in Table 2. In Table 2, LSc is listed. My guess is that it is the same parameter. However, the reader should not guess something, because the reader should know it unequivocally from the content of the article. This is not the case here.
AU: we modified as suggested
The abbreviation (fig. 1a) on the line: 310 should be written with a capital letter, i.e. Fig. 1a. The same remark applies to lines 368, 387, and 405 as well as 406.
AU: we modified as suggested
If the Authors already use the term versus (I don't know why it is italicized?), It is worth sticking to one way of writing. For example, on line 321 it says vs and on line 323 versus, so it's a good idea to standardize the spelling of this word. The same remark applies to lines 482 and 487.
AU: we modified as suggested.
If the Authors titled subchapter 3.5 GLM analyzes, then such a title is difficult for me to identify, because at the moment I do not remember what the acronym GLM stands for. Of course, I can find the full GLM name in the earlier text, which is a significant difficulty and a source of unnecessary distraction while reading the text. Therefore, I suggest that you include the full name of GLM in the title of subchapter 3.5. Even more so, in the entire text of the article I did not find what GLM means, but only found the wording "GLM procedures" (lines: 142-143), which I do not know what it means, i.e. what procedure it is about. It can be assumed that the reader should know what the GLM procedure means, but in my opinion this is a wrong assumption.
AU: we modified as suggested.
I think that one of the practical conclusions resulting from the research is paying attention to the differentiation of the milk flow rate from each quarter and the milking time of each quarter, which translates into the improvement of the operation of milking machines with the automatic milking remover function. These are apparatuses used in classic milking parlours, therefore the results of the authors' research may be useful for the development of research on these milking apparatuses. It is worth writing about it at the end of the article.
AU: as far as we know, at least in Italian dairy sector, milking systems (no AMS) with measurement at quarter level are not available. For this reason, at the moment, unfortunately, these results are only applicable for AMS
Duplicate publication numbering must be removed in References.
AU: we re-write the references section
Reviewer 2 Report
The Introduction chapter is very limited in the references cited. There are only four of them. It is true that the authors took up various issues related to milking automation in the Introduction, but in my opinion they require more complete documentation in the form of quoting specialist literature. The literature in the field of research undertaken is very rich and it is worth using it in the form of citations. For example, the health issues of cows milked with a robotic milking and classic milking systems can be developed by citing selected studies, for example, "Assessment of dairy cow herd indices associated with different milking systems".
In my opinion, after the review of the state of knowledge in the Introduction, it would be worth formulating a research problem. The research problem should result from a knowledge review indicating an insufficient state of knowledge in the area in question. This insufficient state of knowledge concerns in a given case - in my opinion - the possibility / reliability of using automatically collected data by AMS to assess health problems of dairy cows. Only the formulation of the research problem should be a premise to identify a gap in the current state of knowledge and to present the purpose of the research. In general, in the case of the formulated research goal, it would be worthwhile to simultaneously indicate a cognitive (scientific) goal and a utilitarian (useful) goal.
On line 65, the authors wrote "(VMS, Delaval, Tumba, Sweden - A AMS)" and on line 66, "AMS, Lely Industries NV, Maassluis, the Netherlands - B AMS)". I am already omitting the fact that the poem with the Dutch company does not have an opening parenthesis at the beginning. However, it would be worthwhile to write "A AMS" and "B AMS" in a different way in the text of the article. Certainly, the authors' intention was to provide in lines 66 and 67 the designations of these two systems with "A AMS" and "B AMS". However, these markings are completely unreadable. I suggest you write a separate sentence that for research purposes, DeLaval's Milking Robots is labeled "A AMS" and Lely's Milking Robots is labeled "B AMS".
It's worth checking it out, but in my opinion, the name of the Swedish manufacturer is written as DeLaval, not Delaval (as the authors of the article wrote). It is therefore worth correcting the provisions of the company name in the article.
In my opinion, the description of the research material is not sufficient. If the research took into account SCC and other data related to the health and risk of cows, it would be advisable to indicate what type of laying bed was used on the farms. How often has the bedding material been replaced or refilled (in the case of straw / sawdust / sand) or cleaned. Were the cows kept in buildings with a loose housing system or a free-stall system? Was the lying area in the lying stalls (if the free-stall system) met the requirements of hygiene and how was this hygiene maintained (the employee removed the impurities?).
I learned from the Materials and Methods chapter that the research included data from four farms, two for each company equipping farms with automatic milking systems. The authors also wrote how many cows the data came from and how much data there is, collected in the period 2016-2019. However, I did not find out how many milking robots were working on each farm. Were they one-stall milking robots? Theoretically, readers should know if they were one-stall AMS, if they are familiar with the companies' offerings. However, probably not everyone needs to know it, and therefore it would be worth confirming in the description in the Materials and Methods chapter. For me, the DeLaval VMS model is clearly identified. However, for the Lely milking robots included in the studies, more details need to be provided. Have there been a Lely Astronaut or A3 or A4 milking robot model on the farms? Please provide these details as they are important for the general description of the research material (study objects).
In the Materials and Methods chapter, the authors wrote that the data came from 720 cows (operated by the A AMS robot) and 283 cows (operated by the B AMS robot). And I would like to know the size of the herd milked with a milking robot on each farm under real conditions. What was the average herd size (± SD - standard deviation) milked with a milking robot on the farm and was it within the required range. For me, such information is the basic information presenting research objects, which in my opinion constitute a key element of description and comparison with other studies, as well as the repeatability of the authors' own research.
If the authors consider in-line and on-line sensors, it would be worth writing where they are exactly in the milking robot installation. Such details are not known to all readers, so - in my opinion - it needs to be described in more detail.
In subsection 2.2 (Statistical analyses) it would be worth mentioning the use of descriptive statistics to present the characteristics of cows / herds / farms. The results of descriptive statistics are presented in subsection 3.1, but it should be mentioned earlier, i.e. in subsection 2.2.
In the commentary to the result data summarized in Table 2, the Authors mentioned LS (line: 167), but such a concept (LS) is not in Table 2. In Table 2, LSc is listed. My guess is that it is the same parameter. However, the reader should not guess something, because the reader should know it unequivocally from the content of the article. This is not the case here.
The abbreviation (fig. 1a) on the line: 310 should be written with a capital letter, i.e. Fig. 1a. The same remark applies to lines 368, 387, and 405 as well as 406.
If the Authors already use the term versus (I don't know why it is italicized?), It is worth sticking to one way of writing. For example, on line 321 it says vs and on line 323 versus, so it's a good idea to standardize the spelling of this word. The same remark applies to lines 482 and 487.
If the Authors titled subchapter 3.5 GLM analyzes, then such a title is difficult for me to identify, because at the moment I do not remember what the acronym GLM stands for. Of course, I can find the full GLM name in the earlier text, which is a significant difficulty and a source of unnecessary distraction while reading the text. Therefore, I suggest that you include the full name of GLM in the title of subchapter 3.5. Even more so, in the entire text of the article I did not find what GLM means, but only found the wording "GLM procedures" (lines: 142-143), which I do not know what it means, i.e. what procedure it is about. It can be assumed that the reader should know what the GLM procedure means, but in my opinion this is a wrong assumption.
I think that one of the practical conclusions resulting from the research is paying attention to the differentiation of the milk flow rate from each quarter and the milking time of each quarter, which translates into the improvement of the operation of milking machines with the automatic milking remover function. These are apparatuses used in classic milking parlours, therefore the results of the authors' research may be useful for the development of research on these milking apparatuses. It is worth writing about it at the end of the article.
Duplicate publication numbering must be removed in References.
Author Response
REV 2
The Introduction chapter is very limited in the references cited. There are only four of them. It is true that the authors took up various issues related to milking automation in the Introduction, but in my opinion they require more complete documentation in the form of quoting specialist literature. The literature in the field of research undertaken is very rich and it is worth using it in the form of citations. For example, the health issues of cows milked with a robotic milking and classic milking systems can be developed by citing selected studies, for example, "Assessment of dairy cow herd indices associated with different milking systems".
AU: thank you for your suggestion, we added some sentences and citations in the introduction section
In my opinion, after the review of the state of knowledge in the Introduction, it would be worth formulating a research problem. The research problem should result from a knowledge review indicating an insufficient state of knowledge in the area in question. This insufficient state of knowledge concerns in a given case - in my opinion - the possibility / reliability of using automatically collected data by AMS to assess health problems of dairy cows. Only the formulation of the research problem should be a premise to identify a gap in the current state of knowledge and to present the purpose of the research. In general, in the case of the formulated research goal, it would be worthwhile to simultaneously indicate a cognitive (scientific) goal and a utilitarian (useful) goal.
AU: we followed your suggestion and modified the end of introduction section.
On line 65, the authors wrote "(VMS, Delaval, Tumba, Sweden - A AMS)" and on line 66, "AMS, Lely Industries NV, Maassluis, the Netherlands - B AMS)". I am already omitting the fact that the poem with the Dutch company does not have an opening parenthesis at the beginning. However, it would be worthwhile to write "A AMS" and "B AMS" in a different way in the text of the article. Certainly, the authors' intention was to provide in lines 66 and 67 the designations of these two systems with "A AMS" and "B AMS". However, these markings are completely unreadable. I suggest you write a separate sentence that for research purposes, DeLaval's Milking Robots is labeled "A AMS" and Lely's Milking Robots is labeled "B AMS".
AU: thank you, we modified as suggested.
It's worth checking it out, but in my opinion, the name of the Swedish manufacturer is written as DeLaval, not Delaval (as the authors of the article wrote). It is therefore worth correcting the provisions of the company name in the article.
AU: thank you, we modified as suggested.
In my opinion, the description of the research material is not sufficient. If the research took into account SCC and other data related to the health and risk of cows, it would be advisable to indicate what type of laying bed was used on the farms. How often has the bedding material been replaced or refilled (in the case of straw / sawdust / sand) or cleaned. Were the cows kept in buildings with a loose housing system or a free-stall system? Was the lying area in the lying stalls (if the free-stall system) met the requirements of hygiene and how was this hygiene maintained (the employee removed the impurities?).
AU: as suggested by the reviewers we added a table (table 1) with main farm characteristics and some details on the housing.
I learned from the Materials and Methods chapter that the research included data from four farms, two for each company equipping farms with automatic milking systems. The authors also wrote how many cows the data came from and how much data there is, collected in the period 2016-2019. However, I did not find out how many milking robots were working on each farm. Were they one-stall milking robots? Theoretically, readers should know if they were one-stall AMS, if they are familiar with the companies' offerings. However, probably not everyone needs to know it, and therefore it would be worth confirming in the description in the Materials and Methods chapter. For me, the DeLaval VMS model is clearly identified. However, for the Lely milking robots included in the studies, more details need to be provided. Have there been a Lely Astronaut or A3 or A4 milking robot model on the farms? Please provide these details as they are important for the general description of the research material (study objects).
In the Materials and Methods chapter, the authors wrote that the data came from 720 cows (operated by the A AMS robot) and 283 cows (operated by the B AMS robot). And I would like to know the size of the herd milked with a milking robot on each farm under real conditions. What was the average herd size (± SD - standard deviation) milked with a milking robot on the farm and was it within the required range. For me, such information is the basic information presenting research objects, which in my opinion constitute a key element of description and comparison with other studies, as well as the repeatability of the authors' own research.
AU: we added this type of information in table 1
If the authors consider in-line and on-line sensors, it would be worth writing where they are exactly in the milking robot installation. Such details are not known to all readers, so - in my opinion - it needs to be described in more detail.
AU: in our opinion the detailed description of the sensors and their positioning risks to lengthen and burden the text and it is not crucial for the aims of the paper. We provided to delete from the manuscript the expression on-line and in-line
In subsection 2.2 (Statistical analyses) it would be worth mentioning the use of descriptive statistics to present the characteristics of cows / herds / farms. The results of descriptive statistics are presented in subsection 3.1, but it should be mentioned earlier, i.e. in subsection 2.2.
AU: If we are not wrong we mentioned these at lines 175-176
In the commentary to the result data summarized in Table 2, the Authors mentioned LS (line: 167), but such a concept (LS) is not in Table 2. In Table 2, LSc is listed. My guess is that it is the same parameter. However, the reader should not guess something, because the reader should know it unequivocally from the content of the article. This is not the case here.
AU: we modified as suggested
The abbreviation (fig. 1a) on the line: 310 should be written with a capital letter, i.e. Fig. 1a. The same remark applies to lines 368, 387, and 405 as well as 406.
AU: we modified as suggested
If the Authors already use the term versus (I don't know why it is italicized?), It is worth sticking to one way of writing. For example, on line 321 it says vs and on line 323 versus, so it's a good idea to standardize the spelling of this word. The same remark applies to lines 482 and 487.
AU: we modified as suggested.
If the Authors titled subchapter 3.5 GLM analyzes, then such a title is difficult for me to identify, because at the moment I do not remember what the acronym GLM stands for. Of course, I can find the full GLM name in the earlier text, which is a significant difficulty and a source of unnecessary distraction while reading the text. Therefore, I suggest that you include the full name of GLM in the title of subchapter 3.5. Even more so, in the entire text of the article I did not find what GLM means, but only found the wording "GLM procedures" (lines: 142-143), which I do not know what it means, i.e. what procedure it is about. It can be assumed that the reader should know what the GLM procedure means, but in my opinion this is a wrong assumption.
AU: we modified as suggested.
I think that one of the practical conclusions resulting from the research is paying attention to the differentiation of the milk flow rate from each quarter and the milking time of each quarter, which translates into the improvement of the operation of milking machines with the automatic milking remover function. These are apparatuses used in classic milking parlours, therefore the results of the authors' research may be useful for the development of research on these milking apparatuses. It is worth writing about it at the end of the article.
AU: as far as we know, at least in Italian dairy sector, milking systems (no AMS) with measurement at quarter level are not available. For this reason, at the moment, unfortunately, these results are only applicable for AMS
Duplicate publication numbering must be removed in References.
AU: we re-write the references section
Round 2
Reviewer 1 Report
The authors responded to the comments adequately. They added text and fixed errors. I no longer have any comments on the text.
Author Response
"I would like to thank the authors for the point-by-point response to the
reviewers' comments. Both reviewers recommend accepting the document in its
current form. I agree that the document has improved after the corrections,
but in my opinion some minor points still need to be corrected. See specific
comments below:
Line 47. Please write “Body Condition Score” in lower case (such as body
live weight).
AU: we modified as suggested
Line 48. The number of daily milkings (between 2.5 and 3) is confusing. The
average daily milking for a herd may be 2.5, but not for an individual animal.
AU: we modified the sentence in ‘With a number of daily milkings at herd level of about 2.5’
Line 80. The expression of “the risk of increasing of milk somatic cell
count” could be replaced by “of increasing milk cell count or of
increasing of somatic cell count in milk”.
AU: we modified as suggested
Line 85. Please replace “Delaval” with “DeLaval”.
AU: we modified as suggested
Line 124. Please add mathematical formula. Is it ∆Yq=(Yq1-¯Yq)/¯Yq x 100
?
AU: yes, the formula is that one. We added it in the text.
Lines 168-170. Authors stated that GLM procedure was used to compare
different effects. It is obvious that for each cow data were collected over
time (probably on different days of lactation). How were the repeated
measurements managed? How was repeated measurments integrated with the
inclusion of stage of lactation as independent variable in the model? "
AU: the variable stage of lactation was included in the model used. We did not include these results because, in our opinion, the paper is already rich of results.

Reviewer 2 Report
Thank you for answering the questions posed in the review and corrections.
Author Response

(The authors gave the same response as above.)
